# TRiC's tricks inhibit huntingtin aggregation

**Sarah H Shahmoradian**[1,2†‡a], **Jesus G Galaz-Montoya**[2†], **Michael F Schmid**[2], **Yao Cong**[2‡b], **Boxue Ma**[2], **Christoph Spiess**[3‡c], **Judith Frydman**[3], **Steven J Ludtke**[2], **Wah Chiu**[1,2]*

[1]Department of Molecular Physiology and Biophysics, Baylor College of Medicine, Houston, United States; [2]National Center for Macromolecular Imaging, and the Verna and Marrs McLean Department of Biochemistry and Molecular Biology, Baylor College of Medicine, Houston, United States; [3]Department of Biology, Stanford University, Stanford, United States

**\*For correspondence:** wah@
bcm.edu

[†]These authors contributed
equally to this work

[‡]**Present address:** [a]Center for
Cellular Imaging and
NanoAnalytics, Department of
Biosystems Science and
Engineering, Structural Biology
and Biophysics, University Basel,
Basel, Switzerland; [b]State Key
Laboratory of Molecular Biology,
Institute of Biochemistry and Cell
Biology, Shanghai Institutes for
Biological Sciences, Shanghai,
China; [c]Department of Antibody
Engineering, Genentech Inc,
South San Francisco, United
States

**Competing interests:** The
authors declare that no
competing interests exist.

**Reviewing editor**: Werner
Kühlbrandt, Max Planck Institute
for Biophysics, Germany

**Abstract** In Huntington's disease, a mutated version of the huntingtin protein leads to cell death. Mutant huntingtin is known to aggregate, a process that can be inhibited by the eukaryotic chaperonin TRiC (TCP1-ring complex) in vitro and in vivo. A structural understanding of the genesis of aggregates and their modulation by cellular chaperones could facilitate the development of therapies but has been hindered by the heterogeneity of amyloid aggregates. Using cryo-electron microscopy (cryoEM) and single particle cryo-electron tomography (SPT) we characterize the growth of fibrillar aggregates of mutant huntingtin exon 1 containing an expanded polyglutamine tract with 51 residues (mhttQ51), and resolve 3-D structures of the chaperonin TRiC interacting with mhttQ51. We find that TRiC caps mhttQ51 fibril tips via the apical domains of its subunits, and also encapsulates smaller mhtt oligomers within its chamber. These two complementary mechanisms provide a structural description for TRiC's inhibition of mhttQ51 aggregation in vitro.

## Introduction

The presence of huntingtin aggregates in the brain correlates strongly with functional deficits in individuals with Huntington's disease (HD) (*DiFiglia et al., 1997*). These aggregates originate from a mutation in the huntingtin gene that causes an expansion of glutamine (Q) repeats in the huntingtin protein (*Zoghbi and Orr, 2009*). This polyglutamine expansion (Q>36) contributes to mutant huntingtin's (mhtt) tendency to aggregate within the cell (*Nekooki-Machida et al., 2009*). Variants of mhtt exon1 containing an expanded polyglutamine (polyQ) tract greater than 36 residues are commonly used as models to investigate mutant huntingtin's properties in vitro and in vivo. Understanding how polyQ aggregation is modulated by cellular factors, including molecular chaperones, might lead to therapeutic strategies to treat polyQ pathogenesis. Some chaperones, such as the eukaryotic chaperonin TRiC, are key regulators of protein aggregation (*Muchowski and Wacker, 2005*; *Liebman and Meredith, 2010*; *Voisine et al., 2010*). In fact, TRiC modulates and suppresses polyglutamine aggregation, including that of mhtt variants having polyQ tracts of different length. TRiCs inhibition of mhtt aggregation has been biochemically characterized for several variants both in vitro and in vivo (*Kitamura et al., 2006*; *Tam et al., 2006*).

TRiC is a ~1 MDa double-ring complex that interacts with ~10% of all cytosolic proteins and is stringently required for the proper folding of many essential proteins (*Yam et al., 2008*). TRiC can assume different conformations in different biochemical states (*Cong et al., 2012*) and can bind substrates like actin and tubulin through different sites, either at its apical tips, which are conformationally flexible in the apo state (*Llorca et al., 1999*), or inside the chamber (*Dekker et al., 2011*; *Muñoz et al., 2011*). TRiC's interactions with polyQ aggregates are also likely to be variable, as seen with other chaperone/substrate complexes (*Kim et al., 2002*).

**eLife digest** Huntington's disease is an inheritable neurodegenerative disorder that typically begins in mid-adulthood. It initially affects muscle coordination and progresses to include psychiatric symptoms and cognitive decline, leading to premature death. The disease is caused by a mutation in the huntingtin gene, which codes for the huntingtin protein, and all individuals who inherit a pathogenic form of the mutant gene will eventually develop the condition.

The huntingtin gene contains a series of repeats of the tri-nucleotide sequence CAG, which encodes for the amino acid glutamine. The number of repeats varies between individuals but if it exceeds 36, the huntingtin protein starts to form aggregates in the brain. Aggregation occurs when soluble protein precursors, known as oligomers, combine to form structures called fibrils, which in turn assemble into larger clusters. This phenomenon also occurs in several other tri-nucleotide diseases, each of which involves a mutated gene with an excess of tri-nucleotide repeats.

Inside cells, proteins called chaperones regulate the folding of other proteins and help to prevent aggregate formation. A chaperone protein known as TRiC, which interacts with approximately 10% of proteins in the cytosol, has been shown to inhibit the aggregation of mutant huntingtin proteins. However, it has not been possible to map the structural interactions between TRiC and huntingtin to date.

Now, Shahmoradian and Galaz-Montoya et al. have used cryo-electron tomography, combined with 3-D mapping and computer-aided reconstruction, to reveal the structure of a molecular complex consisting of TRiC and a pathogenic mutant huntingtin protein containing 51 CAG repeats. By imaging this system at different time points during the aggregation of mutant huntingtin, it was possible to characterize how the aggregates changed over time. They found that their shape differs in the presence and absence of TRiC, and that the chaperone interacts both with soluble huntingtin molecules—sequestering them so that they cannot join together—and with the tips of fibrils, preventing them from growing longer.

By providing the first direct demonstration of how TRiC inhibits the aggregation of mutant huntingtin, the results of Shahmoradian and Galaz-Montoya et al. could aid in the design of TRiC-based drugs to be used in the treatment of Huntington's disease.

There is no existing structural model for how TRiC suppresses mhtt aggregation, and most structural biology techniques are not suited to characterize conformationally heterogeneous specimens such as mhtt aggregates in complex with TRiC. However, cryo-electron microscopy (cryoEM) can preserve specimens in close-to-solution conformations because the samples remain frozen-hydrated without chemical fixation or staining during the entire microscopy session (*Dubochet et al., 1988*). Additionally, cryo-electron tomography (cryoET) allows for visualization of entire protein aggregates in 3-D. Furthermore, 3-D subtomogram processing approaches (*Walz et al., 1997*; *Böhm et al., 2000*; *Bartesaghi et al., 2008*; *Förster et al., 2008*; *Schmid and Booth, 2008*), also known as single particle tomography (SPT), have facilitated the determination of the structure of heterogeneous specimens by classifying and averaging subvolumes containing a 3-D view of individual macromolecular complexes, potentially with different conformations and compositions. Indeed, SPT is an emerging technique in structural biology that can resolve the structure of heterogeneous macromolecular complexes at nanometer resolution.

## Results

### Mutant huntingtin aggregates display different morphologies in the absence and presence of TRiC

We imaged mhtt exon1 containing 51 glutamine repeats (mhttQ51) ('Materials and methods' under 'In vitro GST-Q51 aggregation assay') using 2-D cryoEM ('Materials and methods' under 'Cryo electron microscopy') to assess the progression of its aggregation over time and how TRiC affects it (*Figure 1*). Amongst the mhtt variants whose susceptibility to TRiC has been previously characterized in vitro and in vivo (*Behrends et al., 2006*; *Tam et al., 2006*), our choice of the mhttQ51 variant was guided by the fact that mutations yielding a polyQ tract longer than ~58Q

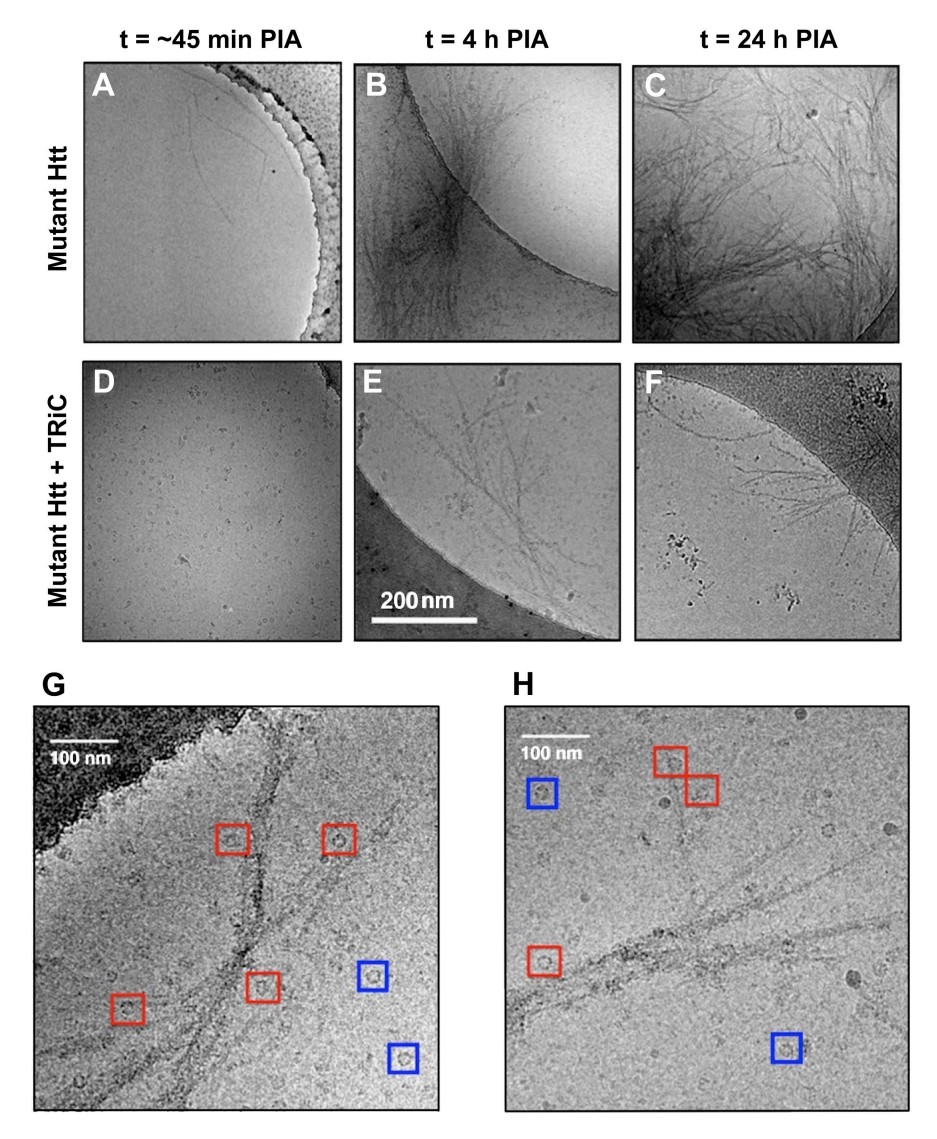

**Figure 1**. The presence of TRiC inhibits the progression of mhttQ51 aggregation. 2-D cryoEM images of (**A**) mhttQ51 at 45 min, (**B**) 4 hr, and (**C**) 24 hr post-initiation of aggregation (PIA), and of mhttQ51 incubated with TRiC and imaged at (**D**) 45 min (**E**) 4 hr and (**F**) 24 hr PIA. (**G**, **H**) Higher magnification images of mhttQ51 + TRiC at 4 hr PIA. Some TRiCs are seemingly attached to mhtt fibrils (red boxes), while others are seemingly freestanding (blue boxes).

are rare in patients (**Myers, 2004**), and N-terminal fragments of similar length (Q53) have been isolated from brain tissue (**Lunkes et al., 2002**). We do not observe any regular repeat in the images of the aggregates, in contrast to more ordered aggregates such as beta amyloid and alpha synuclein (**Lührs et al., 2005**; **Trexler and Rhoades, 2010**). The heterogeneity of mhttQ51 fibrils precluded us from resolving their structure in detail since all current structural techniques require some underlying homogeneity in fibrillar structures to achieve high resolution.

At 45 min post-initiation of aggregation (PIA) in the absence of TRiC, we see a few isolated mhttQ51 fibrils (**Figure 1A**). Even at early time points, mhttQ51 fibers exhibit extensive heterogeneity, varying both in length and width, with the tips appearing more tapered than the bulk of the fiber. By a process that is not yet fully understood, thin fibrils progress to form thicker fibers, and eventually form branched-out aggregates or 'sheaves' with a dense central core by 4 hr PIA (**Figure 1B**). At longer incubation times, we observe a dramatic increase in the number and thickness of such sheaves, characterized as

large fiber bundles densely packed in their central regions and splayed out at both ends. The number of sheaves and their thickness appears to plateau at 24 hr PIA (*Figure 1C*).

On the other hand, in the presence of TRiC, not even thin mhttQ51 fibrils are visible at early time points (45 min PIA; *Figure 1D*). At 4 hr and 24 hr PIA (*Figure 1E,F*), sparse fibrils and a few small sheaves are present, but there are no aggregates as large as those seen in the absence of TRiC at the same time points (*Figure 1B,C*). TRiC does not completely decorate mhttQ51 fibrils, but rather binds at discrete locations, which are often discernible fibril tips (*Figure 1G–H*). In a 2-D cryoEM image, the entire volume of the actual specimen is projected onto a plane; therefore, some TRiCs seemingly close to a fibril in 2-D might be floating above or below it and not bound at all. Since 2-D cryoEM images cannot unambiguously show whether TRiC is directly bound to mhttQ5, we performed cryoET experiments.

We collected and reconstructed 20 tilt series of mhttQ51 incubated with TRiC at 4 hr PIA ('Materials and methods' under 'Cryo electron tomography and tomogram annotation'). We also reconstructed tilt series of TRiC in the absence of mhttQ51 as a control. *Figure 2A* is a slice extracted from one of our mhtt + TRiC tomograms, low-pass filtered for visualization. *Figure 2B* is a re-projection of the entire tomogram (*Video 1*) and *Figure 2C* a manual 3-D annotation of its features, which shows an aggregate of mhttQ51 fibrils in yellow, freestanding TRiC particles in blue, and putatively fibril-bound TRiCs in magenta. As in our 2-D images, our 3-D tomograms also show that TRiC does not fully decorate mhtt fibrils, but rather seems to bind at discrete locations, which often are discernible fibril tips. Our visual estimates from these 3-D tomograms indicate that 37% (120 of 326) of fibril-bound TRiCs locate to obvious and clearly discernible tips. Conversely, about half of the distinguishable fibril tips seem clearly capped by TRiC, but many tips (capped or uncapped) might be obscured by the dense tangles, or may exist as short stubs budding from the side of thicker fibers, which might not be discernible. In fact, given the low signal to noise ratio (S/N) of individual tomograms, their inherently anisotropic resolution, and 'missing wedge' distortions present in all cryoET data, it is challenging to visually discern all TRiC molecules and mhtt fibril tips that might be present in our crowded fibrillar tangles. In addition, some fibril-bound TRiCs might be bound to fibril tips barely 'budding out' from the sides of fibril bundles, too short to be discerned. Therefore, the number of TRiCs that locate to tips is likely to be underestimated when counting only those that are on visually discernible tips.

In both our 2-D images (*Figure 1G–H*) and 3-D tomograms (*Figure 2*, *Video 1*), we identified two distinct populations of particles similar in size and shape to TRiC. Particles in one population seem to be attached to mhttQ51 fibrils ('fibril-bound TRiC'), while those in the other do not ('freestanding TRiC').

To circumvent some of the difficulties presented by the structural heterogeneity of our specimens and to improve the resolvability of features, we extracted subtomograms from the fibril-bound and freestanding TRiC populations and processed them separately to obtain 3-D averages. We developed new algorithms recently available in EMAN2 (*http://blake.bcm.edu/emanwiki/SPT/Spt*) and used them for all our SPT processing.

## Analysis of fibril-bound TRiC suggests an end-on interaction with mhttQ51 fibrils

Given TRiC's aggressive inhibitory effect on mhtt aggregation and the tendency of mhtt aggregates to localize to the carbon instead of the ice in cryoEM grid holes, finding sizable fibrillar aggregates in complex with TRiC proved challenging. We processed a total of 326 fibril-bound TRiC subvolumes extracted from our tomograms using a completely bias-free subvolume alignment and averaging strategy ('Materials and methods' under 'Single particle tomography [SPT]'). This represented about one quarter of 1216 TRiC subvolumes extracted from all our mhtt-TRiC tomograms combined. *Figure 3* shows examples of fibril-bound TRiC averages using subvolumes from separate tomograms. Although the heterogeneity of our fibril-bound TRiC sets precluded them from converging to a single structure and thwarted the determination of specific binding sites between TRiC and mhttQ51 fibrils, all our datasets yielded consistent barrel-shaped averages clearly resembling TRiC, with extra mass at the ends of the barrel. These extra densities are attributable to conformational averages of heterogeneous mhttQ51 fibrils. A few averages suggest a mode in which the fibrils might bind predominantly to the tip of a single CCT subunit, while others show mass splayed across the ring, making additional contacts.

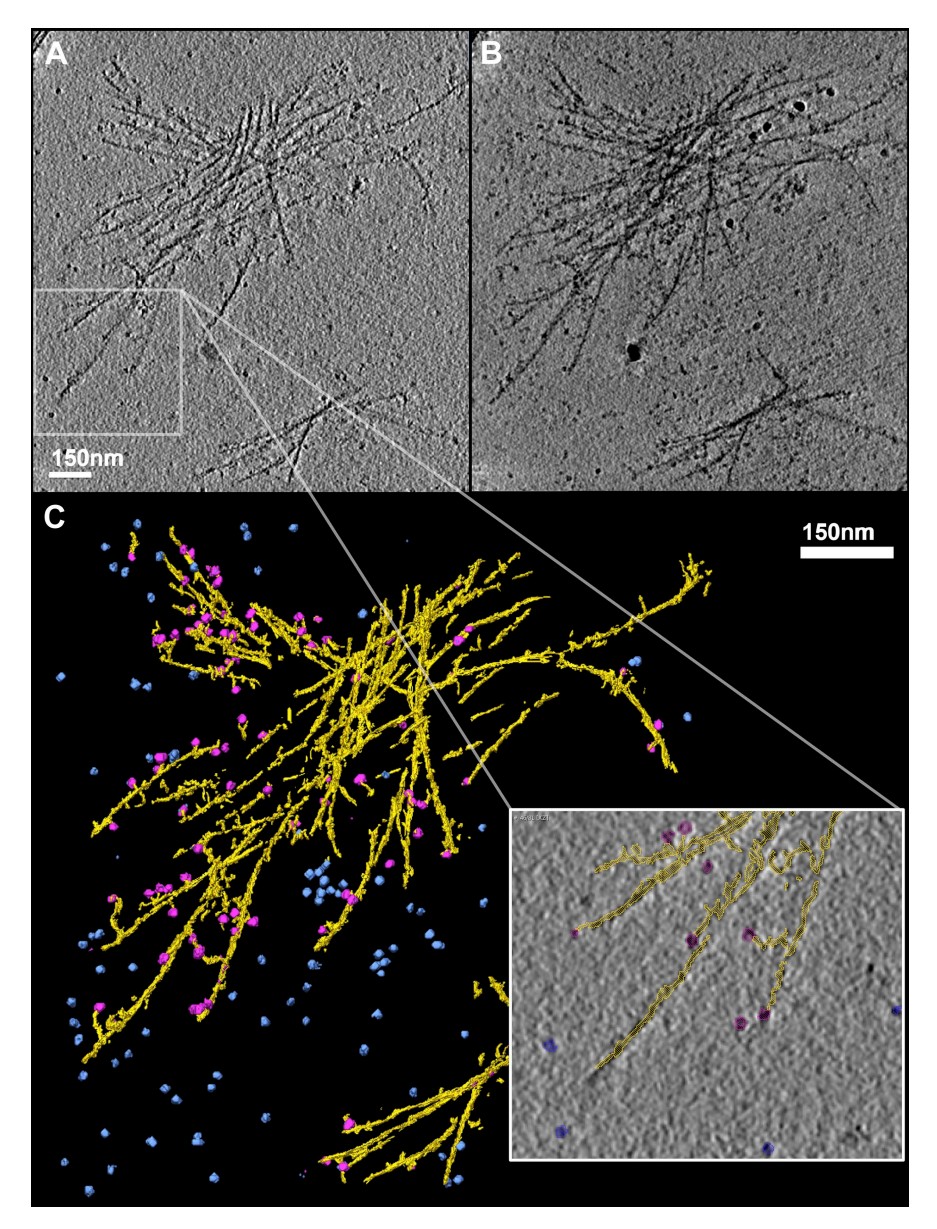

**Figure 2**. TRiC caps the tips of mhttQ51 fibrils. (**A**) 2-D slice through a tomogram of TRiC incubated with mhttQ51 imaged at 4 hr post initiation of aggregation. (**B**) 2-D reprojection of the entire tomogram and (**C**) 3-D annotation of its features, showing mhtt fibrils in yellow, freestanding TRiCs in blue, and fibril-bound TRiCs in magenta. Zoomed-in field shows fibril-bound TRiC localizing mostly to discernible tips of mhtt fibrils (seemingly detached magenta TRiC is attached to a fibril on another z-slice of the tomogram).

## Analysis of freestanding TRiC shows extra density within its central chamber

We also sought to understand whether, and how, freestanding TRiC might interact with smaller mhttQ51 oligomers. We extracted a total of 890 freestanding TRiC subvolumes from our tomograms and performed SPT (i.e., subvolume alignment, classification, and averaging). These were about three quarters of the 1216 TRiC subvolumes extracted from all our mhtt-TRiC tomograms combined. When processing the subvolumes from each tomogram separately, SPT averages were consistent with the results from the combined data set, indicating a high level of self-consistency in the results. Our statistical analyses ('Materials and methods' under 'Criteria to discriminate between cavity-empty

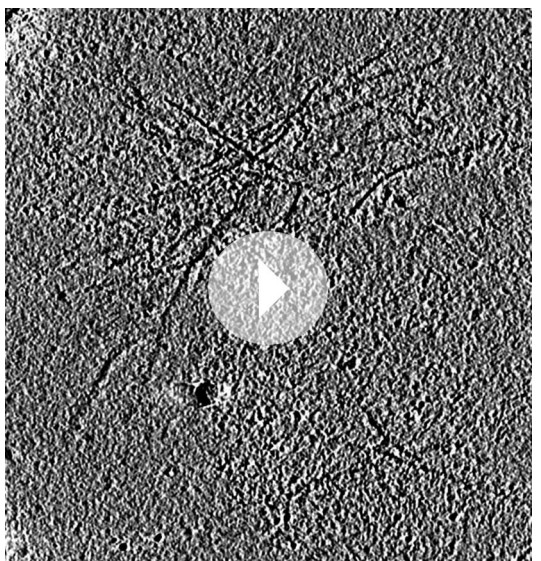

**Video 1**. Annotation of 3-D tomogram of mhtt aggregate in the presence of TRiC.

and cavity-occupied freestanding TRIC', *Figure 4*) suggest that, while the freestanding TRiC subvolumes exhibit a high but continuous level of variability (*Figure 4A*), they can be divided into two statistically separable populations (*Figure 4B*). These populations appear to correspond to cavity-empty and cavity-occupied TRiC.

The final cavity-empty TRiC average from the combined dataset highly resembles apo-TRiC and was produced using the same bias-free approach applied to fibril-bound TRiC ('Materials and methods' under 'Single particle tomography [SPT]'). The average is pseudo eightfold symmetrical, barrel-shaped, and hollow (*Figure 5A*), and accounts for almost one third of the entire 1216 TRiC subvolumes extracted from all of our mhtt-TRiC tomograms combined.

Applying the same bias-free approach to the cavity-occupied TRiC set (also about one third of the entire 1216 TRiC subvolumes) produced an average with significant extra density lining the interior of the chaperonin's cavity, attributable to a poorly localized substrate (*Figure 5B*).

In order to validate our results for freestanding TRiC in the presence of mhtt, we performed control experiments using TRiC in the absence of mhtt (*Figure 6*). When this data was processed identically, we could not identify any subset with significant extra density inside the cavity. This strongly suggests that the large encapsulated density seen in our mhtt-TRiC data derives from the incubation of TRiC with mhtt, rather than residual substrate remaining from the TRiC purification process. While we do not have an experimental assay at this time that can conclusively identify the internal density as mhtt, all other answers seem unlikely.

Computing the Fourier shell correlation (FSC) between the cavity-empty and cavity-occupied TRiC averages gives a value of ~52 Å using the FSC = 0.333 threshold. As these maps are not expected to be identical, this test gives a measure of self-consistency and provides the minimum resolution achieved in the two maps independently. This threshold is appropriate when the two maps are being compared, but not averaged, as a 0.5 threshold is appropriate when a map is being compared to a high-resolution reference (*Rosenthal and Henderson, 2003*). Comparing the maps to the published apo-TRiC structure (*Cong et al., 2012*) using the FSC = 0.5 cutoff ('Materials and methods' under 'Assessment of the quality of bias-free freestanding TRiC averages'; *Figure 7*), yields similar values, ~48 Å and ~46 Å for cavity-occupied and cavity-empty TRiC respectively.

To further localize the interior density in the cavity-occupied TRiC data set, we adopted a hollow-template-guided alignment strategy ('Materials and methods' under 'Localization of density within cavity-occupied TRiC SPT average'). This additional analysis results in a TRiC average with a large density asymmetrically located within the cavity (*Figure 8A*). The interior mass is clearly seen in a difference map between the cavity-empty and the cavity-occupied averages (*Figure 8B*). Of note, the template was completely hollow, and therefore model bias cannot account for the extra density we see inside. The TRiC part of this cavity-occupied average is also pseudo eightfold symmetric (*Figure 8A*) and the dip in the rotational correlation plot (or 'u-shaped' path of the overall curves) can be explained by the presence of internal mass, localized to one side of the cavity when viewed end-on ('Materials and methods' under 'Localization of density within cavity-occupied TRiC SPT average', 'Estimation of the size of mhttQ51 oligomers encapsulated by TRiC').

## Discussion

Our study of mhtt fibrillogenesis in the absence and presence of TRiC provides new insights into how this chaperonin inhibits mhttQ51 aggregation. Our aggregation and suppression of aggregation reactions followed well characterized conditions, previously described using filter trap assays that detect mhttQ51 amyloid aggregates (*Tam et al., 2006*). The mhtt concentration in our experiments was

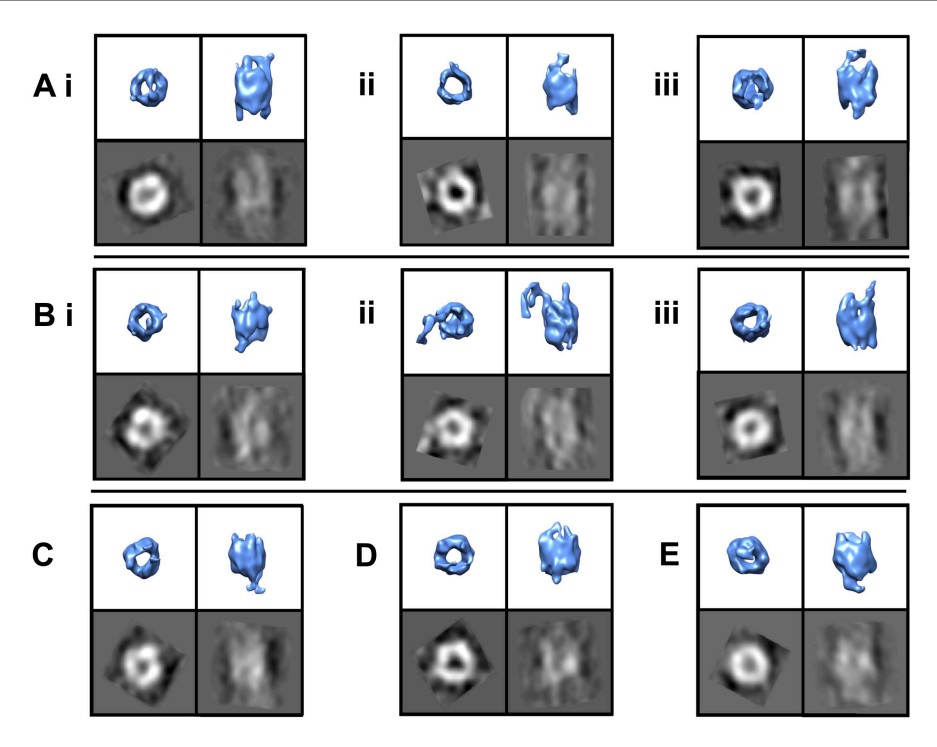

**Figure 3**. TRiC's apical tips bind mhtt-fibrils. SPT symmetry-free and model-free averages of fibril-bound TRiC reveal extra density at the chaperonin's apical tips. Each panel shows an end-on and a side view isosurface for a 3-D average of TRiC in complex with mhttQ51 fibrils, and corresponding 2-D projections (black/white). Independent sets from single tomograms are labeled **A**, **B**, **C**, **D** and **E**. Within each, the numbering (i, ii and iii) indicates mutually exclusive averages. The numbers of subvolumes contributing to each average are (**A**.i.) 7, (**A**.ii.) 10, (**A**.iii.) 13, (**B**.i.) 7, (**B**.ii.) 10, (**B**.iii.) 5, (**C**) 12, (**D**) 8 and (**E**) 12. All the averages show overall TRiC-like morphology with extra densities protruding beyond the apical domains of the chaperonin.

chosen to allow mhtt aggregation to occur in a reasonably short time in the absence of TRiC (*Scherzinger et al., 1999*), and the ratio of TRiC to mhtt was chosen to yield initial-stage fibril aggregates and yet thwart large-aggregate growth (*Behrends et al., 2006*). It is worth noting however that the TRiC:mhtt ratio under normal physiological conditions has not been carefully explored. Nonetheless, increasing cellular levels of TRiC has a beneficial effect in suppressing mhtt toxicity and aggregation (*Behrends et al., 2006*; *Kitamura et al., 2006*; *Tam et al., 2006*).

In our 2-D cryoEM images, the formation of groups of sheaves as seen in *Figure 1C* could correspond to the large inclusions observed in cells expressing mhtt. On the other hand, the smaller aggregates (*Figures 1E,F and 3*), which are not detectable by filter-trap assays, could correspond to the multiple, smaller foci seen in cells overexpressing TRiC or some of its subunits (*Behrends et al., 2006*; *Tam et al., 2006*). Importantly, overexpression of TRiC is known to suppress mhtt toxicity in a variety of systems (*Behrends et al., 2006*; *Kitamura et al., 2006*; *Tam et al., 2006*) and thus our observations provide a structural basis for the observed effect of TRiC.

We envision the progression of mhttQ51 aggregation as a process whereby, after the nucleation stage, small and thin irregular fibrils with exposed hydrophobic moieties ('fibril tips') drive the growth of the aggregates into large sheaves. TRiC is known to bind hydrophobic motifs via its apical substrate-binding domains (*Spiess et al., 2006*), and has also been shown to crosslink with the N-terminal N17 motif of mhtt, which drives mhtt oligomerization (*Tam et al., 2009*). It is therefore likely that the binding we demonstrate here between mhttQ51 fibrils and TRiC (*Figures 1–3*), which thwarts the progression of fibers into sheaves, may involve mhttQ51 hydrophobic moieties that include the N17 domain. Interestingly, this domain has been proposed to function as a 'molecular switch' (*Greiner and Yang, 2011*) controlled by multiple post-translational modifications that can determine the fate of mhtt exon-1 either by rendering it innocuous or enhancing its pathogenicity.

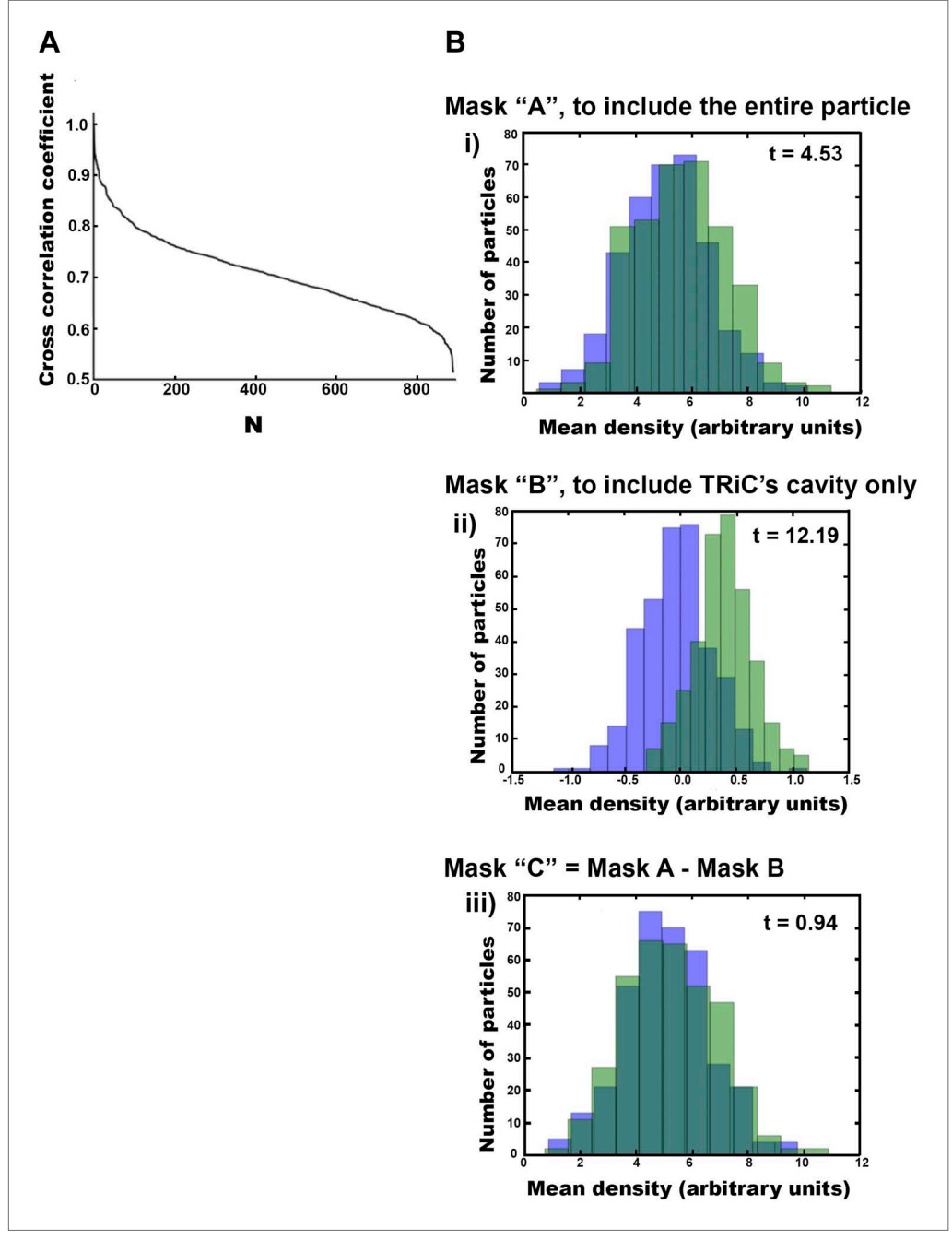

**Figure 4**. Freestanding TRiCs are classifiable into cavity-empty and cavity-occupied. (**A**) Ranked correlation scores for freestanding TRiCs against a hollow TRiC reference yield a continuous fast-decaying plot, suggesting structural heterogeneity. (**B**) Mean-density distributions for cavity-empty TRiC (blue) and cavity-occupied TRiC (green) sets, masking the particles to (i) include all the density in each particle, (ii) include the cavity of each molecule only, and (iii) include TRiC density but exclude the cavity. Corresponding *t*-scores for the difference (or shift) between the blue and green distributions are shown for each masking condition.

By virtue of its interaction with the N17 domain, TRiC might provide a protective effect against deleterious post-translational modifications.

Although we do not observe a unique structure for fibril-bound TRiC, we demonstrate that TRiC binds mhtt fibrils via its apical tips to cap them. This fibril-capping mechanism could explain why inhibition of aggregation is observed in vitro at substoichiometric ratios of TRiC to mhtt. It may also

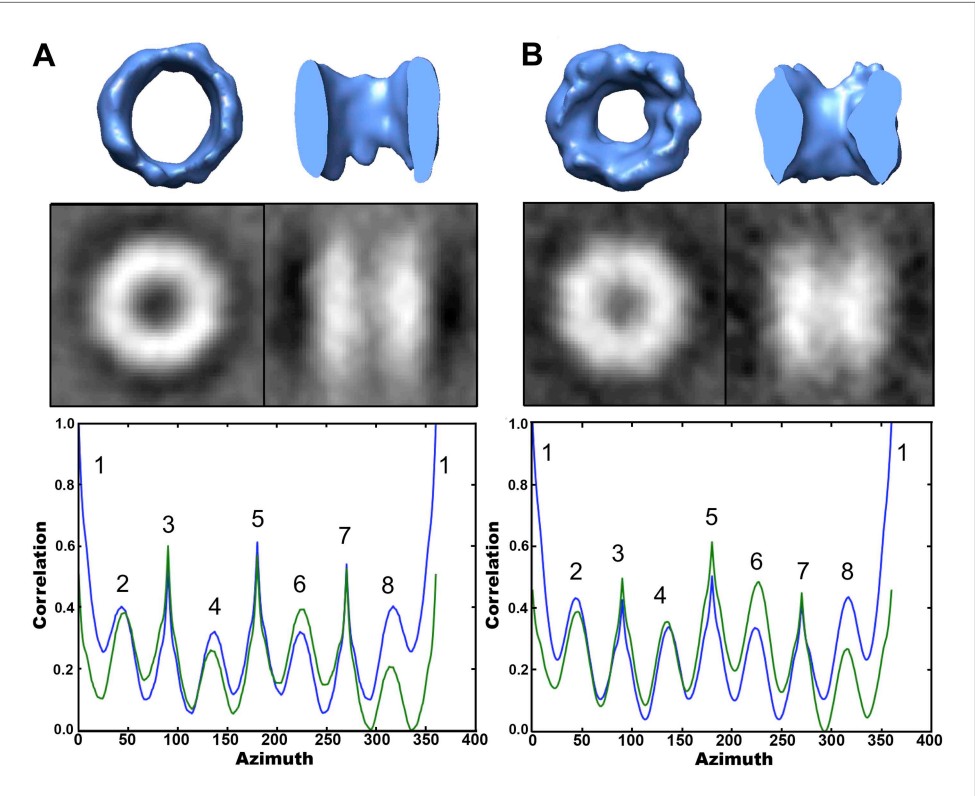

**Figure 5**. A subpopulation of freestanding TRiC contains intra-cavity extra density. SPT symmetry-free and template-free processing of freestanding TRiC demonstrates pseudo-eightfold symmetry in an average of (**A**) 356 cavity-empty TRiCs and (**B**) 356 cavity-occupied TRiCs. 2-D projections for each average are shown in black/white. The plots show volumetric rotational correlation analyses to test for C8 (blue) and D8 (green) symmetry in each average. The cavity size in B is significantly reduced, due to internalized density.

explain why mhtt aggregates look different in vivo when TRiC is overexpressed, and become SDS soluble (*Tam et al., 2006*).

It is conceivable that TRiC might bind fibrils through one of its ends while the binding sites at the other end remain available to snatch mhttQ51 oligomers lurking in their proximity. On the other hand, internalization of smaller mhttQ51 oligomers would also contribute to inhibition of sheaf formation by decreasing the effective concentration of oligomers in the reaction mixture. Our freestanding TRiC averages, free from template and symmetry bias (*Figure 5*), reveal a subpopulation with extra density inside the chaperonin's cavity. Although other substrates have also been shown to bind deep inside the cavity of apo-TRiC (*Muñoz et al., 2011*), the finding that mhtt exon-1 might also be encapsulated is unprecedented. While it is conceivable for native substrates to be copurified with apo-TRiC, their presence within the chaperonin's cavity is rare. For instance, our controls show that extra density is only detectable in ~10% of the particles in the absence of added mhtt. On the other hand, we see significant extra density in ~40–50% of the freestanding TRiC particles from our TRiC + mhtt tomograms. Furthermore, previous studies have demonstrated that TRiC does not bind GST (*Feldman et al., 2003*). Therefore, it is unlikely that either native substrates or the GST moiety that is cleaved from mhttQ51 to initiate aggregation could account for the internalized mass we observe (*Figures 5B and 8*). Rather, this density likely represents mhtt.

Given the modest resolution of our cavity-occupied average, it is challenging to measure the encapsulated mass. However, our estimates suggest an average size of ~60 kDa ('Materials and methods' under 'Estimation of the size of mhttQ51 oligomers encapsulated by TRiC'), which would correspond to an oligomer of about six mhttQ51 exon1 monomers. The mhttQ51 oligomers that might be encapsulated by TRiC could be analogous to precursor forms of the 100–230 kDa toxic oligomers previously reported (*Behrends et al., 2006*). Furthermore, it is possible that the

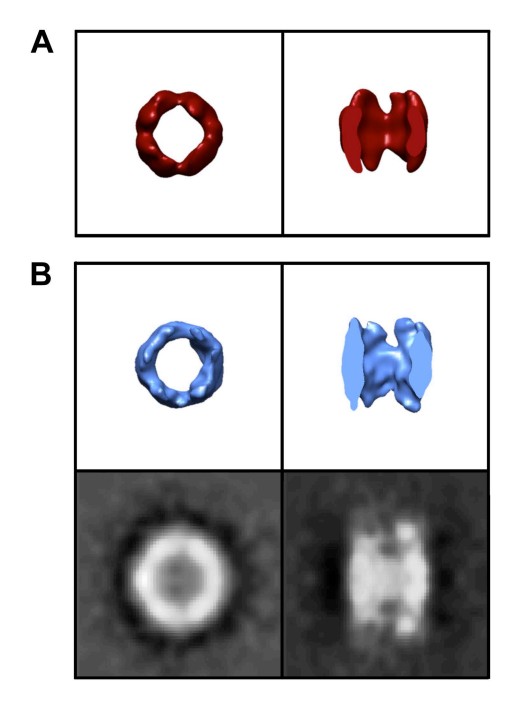

**Figure 6**. Control TRiC in the absence of mhtt. End-on and cut-away views for (**A**) the published cryoEM structure of apo-TRiC (EMDB 1960), low-pass filtered to match the resolution of freestanding TRiCs, and (**B**) model-free SPT TRiC average of 208 subvolumes from TRiC-only tomograms or 'control TRiC' (no mhtt added). Corresponding projections for the control are shown.

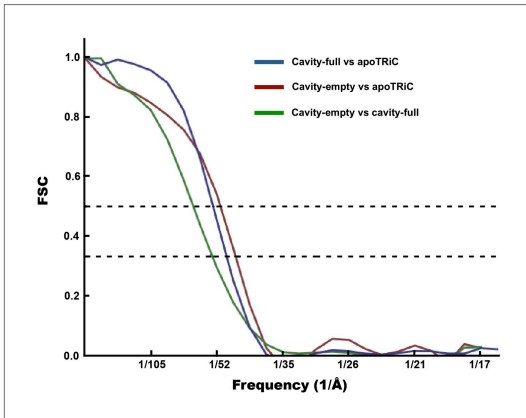

**Figure 7**. FSC curves for freestanding TRiC averages. FSC of cavity-empty TRiC (maroon) and cavity-occupied TRiC (blue) with the published structure of apo-TRiC. The FSC of cavity-empty TRiC with cavity-occupied TRiC is shown in green.

extra density we see inside TRiC (**Figures 5B and 8**) corresponds to an average of encapsulated mhtt oligomers of different sizes and conformations.

While previous biochemical studies showed the inhibitory effect of TRiC on mhttQ51 aggregation, the structural basis of this process was not explained. The compositional and conformational heterogeneity of such a specimen precludes resolving nanometer resolution structures from it with any technique in structural biology other than SPT. Indeed, our approach demonstrates TRiC's association with mhttQ51 fibrillar aggregates, and our results suggest TRiC might also sequester small mhttQ51 oligomers to effectively inhibit mhtt aggregation (**Figure 9, Video 1**). This novel mechanism contrasts with those typically proposed for the action of chaperones in suppressing aggregation of misfolded proteins (**Hendrick and Hartl, 1995**; **Young et al., 2004**) by binding monomeric forms. However, since our analysis lacks the sensitivity to distinguish TRiCs that might be bound to one mhtt monomer only, we cannot rule out that such a binding mode may co-exist with the ones presented here.

While our observation that TRiC can bind substrates via its apical domains is consistent with the current understanding of TRiC–substrate interactions, our data is unprecedented in demonstrating the direct binding of fibrils (a substrate too large to be internalized). The suggestion that TRiC might cap fibril growth as well as internalize smaller oligomers indicates a broad scope for this chaperonin's action on protein aggregates. The ability of TRiC to recognize and handle both types of proposed mhtt toxic species observed in Huntington's disease (**DiFiglia et al., 1997**; **Arrasate and Finkbeiner, 2011**; **Nucifora et al., 2012**), namely oligomers and fibrillar aggregates, could account for the reduced toxicity of mhtQ51 and other mhtt species when TRiC is overexpressed in cells (**Behrends et al., 2006**; **Tam et al., 2006**). TRiC's interaction with mhtt aggregates might render them more amenable to processing by cellular pathways of quality control, such as autophagy and proteolytic degradation by the proteasome (**Sarkar et al., 2009**).

Since mhtt oligomers and fibrillar aggregates are strongly implicated in the pathogenesis of Huntington's disease (**Sathasivam et al., 2010**), the interactions that we describe between TRiC and mhttQ51 provide a structural mechanism for the chaperonin's aggressive inhibition of mhtt aggregation in vitro and suggest a potential for TRiC-inspired therapeutics (**Sontag et al., 2013**).

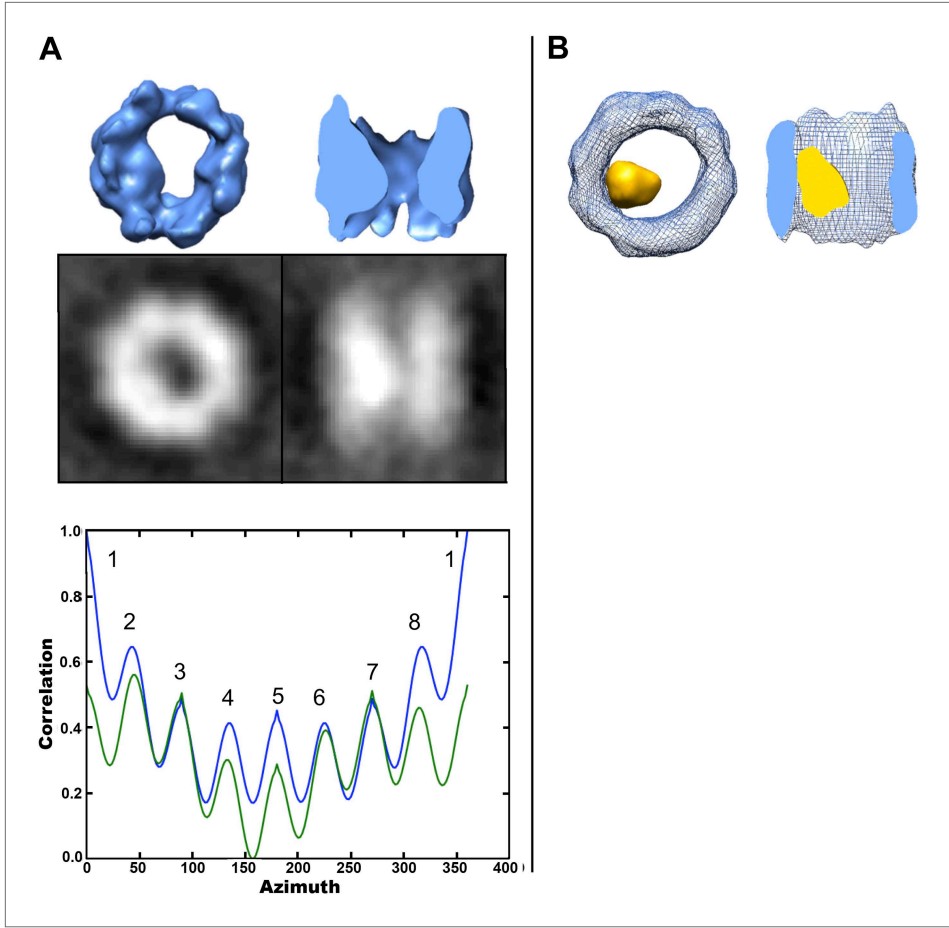

**Figure 8**. Localized intra-cavity extra density within TRiC. (**A**) Hollow-template-guided average of 356 cavity-occupied TRiCs, 2-D projections (black/white) and rotational correlation analysis to test for C8 (blue) and D8 (green) symmetry, showing a 'dip' in self-correlation at azimuth equal to 180° due to the presence of localized inner mass. (**B**) End-on (left) and cut-away side (right) views of the superimposition of the cavity-empty TRiC average (blue wire) from Figure 5A, and the difference map (yellow) between it and the cavity-occupied average in (**A**).

## Materials and methods

### In vitro GST-Q51 aggregation assay

Aggregation of GST-huntingtin exon 1 fusion proteins with 51 glutamine repeats (GST-mhttQ51) at a concentration of 6 µM was initiated in vitro by adding AcTEV protease (Invitrogen), both alone and in presence of 1.2 µM TRiC at a t = 0 hr (*Tam et al., 2006*) in TEV Buffer (Invitrogen, Grand Island, NY). Samples were subjected to spin-column equilibration using Bio-Spin Columns with Bio-Gel P-30 (Bio-Rad). Samples were incubated at 30°C and aliquots vitrified at t= 0 hr, 0.75 hr, 4 hr, and 24 hr.

### Cryo electron microscopy

All aliquots were applied to 200-mesh holey carbon Quantifoil copper grids (rinsed in PBS, Biological Industries) and plunge-frozen into liquid ethane at liquid nitrogen temperature (unfixed, unstained), as described (*Dubochet et al., 1988*). 2-D images were collected using the standard minimum dose system on a 200KV JEOL electron microscope (JEM 2100, LaB$_6$ gun), with a sampling of 2.86 Å/pixel, a target underfocus of 4 µm and a total dose of ~20 e/Å$^2$.

### Cryo electron tomography and tomogram annotation

We collected 20 tilt series of mhttQ51 + TRiC at t = 4 hr using SerialEM (*Mastronarde, 2005*) on a JEM2100 electron microscope equipped with a LaB$_6$ gun operating at 200 kV, from −60° to 60°, at 2° increments,

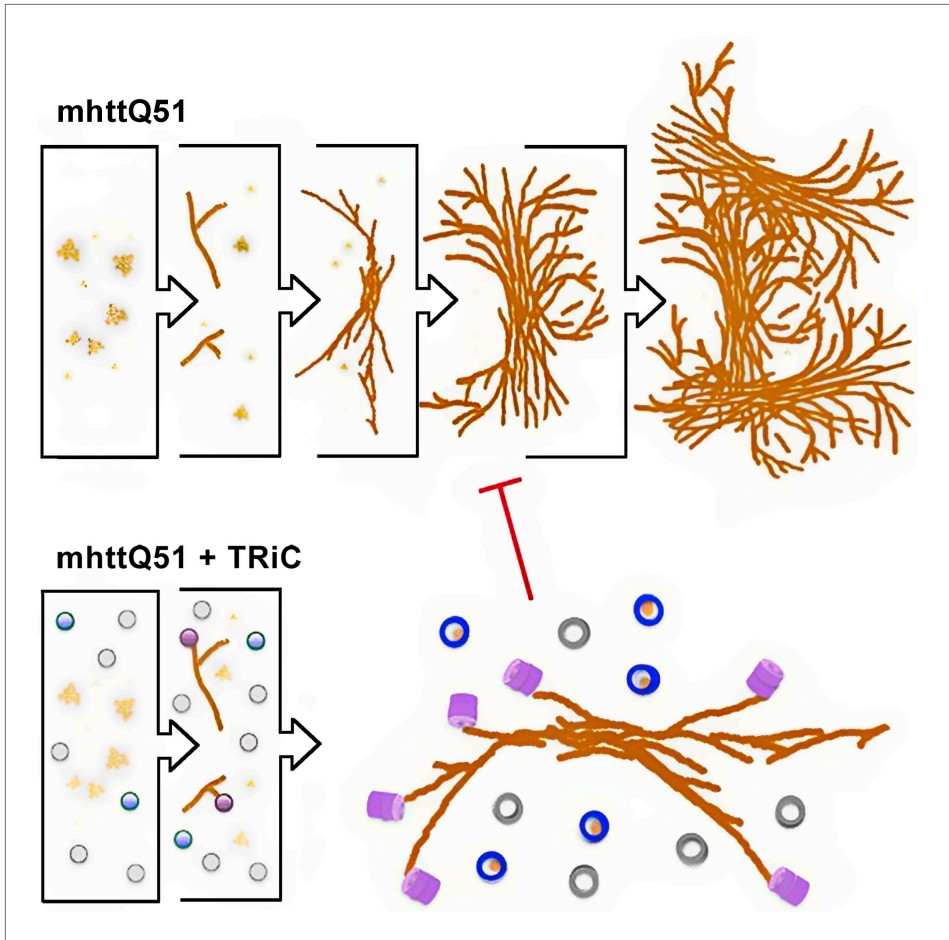

**Figure 9**. Model for the progression of mhttQ51 aggregation in the absence and presence of TRiC. In the absence of TRiC (top row), mhttQ51 (orange) aggregation progresses from monomers and oligomers into fibrils and large sheaves. On the other hand, the presence of TRiC (bottom row) prevents the progression of mhtt fibrils into the large bundled 'sheaf' form by capping (purple) fibril tips and encapsulating (blue) mhtt oligomers. TRiCs that are not bound to mhtt are shown in gray.

target underfocus of 5 μm, sampling of 4.4 Å/pixel and a cumulative dose of ~62 e/Å². 6 of the 20 tilt series appeared appropriate for tomographic reconstruction and were reconstructed into tomograms using IMOD (*Kremer et al., 1996*). The 14 unused tilt series either displayed charging, thick ice, or suboptimal sample distribution. MhttQ51 fibrils were annotated in yellow, while fibril-bound TRiC were annotated in magenta and freestanding TRiC in blue using AVIZO 6.1 software (Visage Imaging GmbH) (*Figure 2C*).

## Single particle tomography (SPT)

All SPT processing and averaging was done with subtomograms extracted from the original raw tomograms (unbinned) using *e2spt_boxer.py* in EMAN2 (*Tang et al., 2007*). To avoid missing wedge bias, we used cross-correlation map normalization (*Schmid and Booth, 2008*). Our bias-free subtomogram processing uses unsupervised hierarchical ascendant classification and averaging (HAC), a technique involving pairwise alignment of sub-tomograms and clustering based on similarity (*Bartesaghi et al., 2008*; *Förster et al., 2008*; *Schmid and Booth, 2008*), without the aid of a reference and without imposing symmetry at any point. For fibril-bound TRiC, we visually inspected the new averages generated in each iteration to find representative examples of structures with extra density at the ends of the chaperonin (*Figure 3*). For each freestanding TRiC population (cavity-empty and cavity-occupied) as well as control TRiCs, we allowed HAC to converge to one final average of all the

particles (*Figure 5A,B*). We noticed significant but poorly localized extra density inside the chamber of the cavity-occupied population.

## Criteria to discriminate between cavity-empty and cavity-occupied freestanding TRIC

We tested our hypothesis that some freestanding TRiCs might be empty while others might contain mhttQ51 oligomers by aligning the particles against a hollow D8 symmetrical apo-TRiC reference (*Booth et al., 2008*) and ranking them according to their correlation with it. Cavity-empty TRiCs would have a higher correlation to the hollow reference than cavity-occupied ones. Indeed, our classification method separated freestanding TRiCs into two groups, and our statistical analyses show that the difference in mass within the cavity between the groups is significant (*Figure 4B*).

When processing the subvolumes in each tomogram, we divided the set into two halves: TRiCs with highest correlation against the reference or 'cavity-empty group', and TRiCs with lowest correlation against the reference or 'cavity-occupied group'. Indeed, the average of the first group was a hollow apo-TRiC-like structure, while the average from the second showed a large but poorly localized mass within the chaperonin's cavity.

The larger set from grouping the freestanding TRiC subtomograms from all our tomograms (890 subtomograms) allowed for a more scrupulous classification, although there is no clear 'cutoff' in the plot of ranked correlation coefficients from aligning the particles against the D8 TRiC reference (*Figure 4A*). To discriminate between cavity-empty and cavity-occupied TRiC, we divided the ranked subtomograms into quintiles (subsets of 178) and averaged the particles in each quintile. The average from each of the two highest quintiles had an empty cavity, while the cavity for each average from the two lowest quintiles contained a large density. The average from the middle quintile was neither clearly empty nor significantly occupied; therefore, we did not subject those subtomograms to further analysis. We made a final cavity-empty set from the two top quintiles and a final cavity-occupied set from the two bottom quintiles, each with 356 subtomograms. That is, each freestanding set (cavity-empty and cavity-occupied) was comprised of 40% of all freestanding TRiCs. In the context of all TRiC subvolumes extracted, each of these sets represented about one third of all particles picked (i.e., 356 of 1216).

We tested the statistical significance of our classification by performing *t*-tests on histograms (*Figure 4B*) that compared the mean density distribution of the cavity-empty group against that of the cavity-occupied group under three separate masking conditions, to examine the density variations across different regions of the subvolumes. The first mask included all the density in each subvolume (mask A), the second mask included only the density in each particle's cavity (mask B), and the third mask was the difference between the first two masks, thus corresponding to TRiC's density in each subvolume (mask C). For each masking condition, we plotted the mean density distribution of each set (cavity-empty and cavity-occupied) as separate histograms in one plot.

The displacement between the cavity-empty and the cavity-occupied histograms using mask A is statistically significant beyond a 99.9% confidence interval (CI), according to its *t*-value. This means that the cavity-empty particles are significantly different from the cavity-occupied particles when the entire density in the subvolumes is considered. The displacement between the histograms using mask B is also statistically significant beyond a 99.9% confidence interval (CI), and the difference is much larger according to the *t*-value. This means that the difference between the particles in each group is very large when only the cavity is considered. Lastly, there was no significant displacement between the histograms using mask C, as confirmed by its low *t*-value, far below the 95% CI value. This means that when the cavity is excluded, the particles in each group are not significantly different. Therefore, our analyses show that the statistically significant differences between the cavity-empty and cavity-occupied TRiC sets are due to the central chamber of the chaperonin rather than the chaperonin itself. This validated our classification and justified further processing the cavity-empty group independently from the cavity-occupied one.

## Assessment of the quality of bias-free freestanding TRiC averages

TRiC has eight highly similar subunits; therefore, if the densities from our tomograms picked as TRiC were in fact TRiC and our methodology averaged them correctly, our maps would display pseudo-eightfold symmetry. Indeed, our rotational correlation plots on the template-free and symmetry-free TRiC averages reveal the presence of pseudo-eightfold symmetry in them (*Figure 5*). We assessed the

resolution of the cavity-empty structure as 46 Å by computing an FSC with the apo-TRiC map (*Cong et al., 2012*) using the FSC = 0.5 threshold, as is appropriate when comparing a map to a much higher resolution structure, as formulated previously (*Rosenthal and Henderson, 2003*) (*Figure 7*).

## Localization of density within cavity-occupied TRiC SPT average

Because TRiC's eight subunits differ the most in their apical tips and their C-termini within the cavity, both containing substrate-binding sites, it is unlikely that any internalized oligomers would bind to all subunits uniformly or line up TRiC's cavity (*Figure 5B*). Therefore, we applied a hollow-template-guided approach to further localize the inner mass within the chamber of the cavity-occupied TRiC average. This consisted of aligning the raw cavity-occupied TRiCs to a hollow D8 symmetric apo-TRiC reference (*Booth et al., 2008*) to generate an initial average. We then seeded iterative refinement of the raw subtomograms using this initial average. During all the iterations, the particles were masked to include only the cavity and were allowed only to visit D8 related orientations. Since the first round of alignment was against the D8 reference, restricting subsequent iterations to visit D8-related positions preserved the initial alignment of the particles against the reference while allowing the internal density to converge within the frame of TRiC after 17 iterations. While this procedure localized the internalized density in our cavity-occupied average, it had little effect on the cavity of the cavity-empty set. Of note, the 'dip' in the overall shape of the rotational correlation plot for the cavity-occupied average with the localized density can be explained by the presence of a large mass inside (*Figure 8A*). Applying the same classification and processing strategy to fibril-bound and control TRiCs as applied to freestanding TRiC did not yield any averages with comparable density inside the cavity.

## Estimation of the size of mhttQ51 oligomers encapsulated by TRiC

The normalized cavity-occupied (*Figure 8A*) and cavity-empty (*Figure 5A*) TRiC averages were aligned to one another, and the power spectrum of one structure was filtered to match that of the other. Then, the cavity-empty average was subtracted from the cavity-occupied average to generate a difference map (*Figure 8B*).

We estimated the mass of the mhttQ51 density internalized by TRiC using three different methods. First, we applied a mask to our final cavity-empty average (*Figure 5A*) that included the TRiC density only, thus excluding the cavity. We calculated the mean density inside this mask and multiplied this value by the volume of the mask. This yielded the final total mass of TRiC (without the cavity) in arbitrary units. Then we applied a mask to the difference map between the final cavity-empty average (*Figure 5A*) and the final cavity-occupied average (*Figure 8A*) that would include the region of the cavity only. We also computed the total mass within this mask, in arbitrary units. Using the actual mass of TRiC (~950 kDa) to scale the masses in arbitrary units, we estimated the mass of the density in the cavity to be ~55 kDa.

For our second method of estimation, we first averaged the density values of the histograms in *Figure 4B*(iii). These histograms correspond to the density distribution of TRiC particles when the cavity is excluded, and therefore the average value in arbitrary units is proportional to TRiC's mass. On the other hand, the histograms in *Figure 4B*(ii) correspond to the density distribution within the cavity of TRiC particles. The average density of the histogram from the cavity-empty group (blue) was considered to be 'background', in arbitrary units. Therefore, we subtracted this value from the average of the cavity-occupied histogram (green). This final value for average density in the cavity region in arbitrary units should be proportional to the average mass of encapsulated oligomers. Knowing the actual mass of TRiC (~950 kDa) allowed for the scaling of masses in arbitrary units, yielding a value of ~65 kDa for the mass within the cavity. Taken together, our two estimates indicate an average size of ~60 kDa for mhttQ51 oligomers encapsulated by TRiC.

Finally, in the rotational correlation plots for the cavity-empty average (*Figure 5A*) and the cavity-occupied average before the substrate was localized (*Figure 5B*), the seven peaks in correlation (excluding the first peak) have a somewhat uniform height. On the other hand, the plot dips at azimuth equal to 180° for the cavity-occupied average (*Figure 8A*). The first peak is bound to be higher in any rotational correlation plot because self-correlation will always be at its maximum when no rotations have occurred (all voxels in the volume correlate well; even those beyond the molecule but within the box). The difference in height (or correlation) between the next two highest peaks (two and eight) with the middle peak (five) in the rotational correlation plot of the cavity-occupied average arises from

the mass of the encapsulated density: when the mass coincides perfectly with its copy, there is high correlation; when it is in opposite alignment to itself, the correlation dips. Therefore, using the mass of TRiC (~950 kDa) to scale this difference indicates a substrate size of ~50 kDA. This is slightly under-estimated, since the difference would ideally be calculated between the first and the middle peaks, but the first peak is unusable due to the perfect correlation of all voxels at the default position before any rotations occur.

## Accession numbers

The Electron Microscopy Data Bank accession numbers for the TRiC-mhttQ51 structures reported in this paper are as follows: For the cavity-empty TRiC average as shown in *Figure 5A*, the EMDB accession number is EMD-5531. For the cavity-occupied TRiC average as shown in *Figure 8A*, the EMDB accession number is EMD-5530. For two of the fibril-bound TRiC averages shown in *Figure 3A*, the EMDB accession numbers are EMD-5532 and EMD-5533.

## Acknowledgements

This research has been supported by the NIH grants PN2EY016525, P41GM103832 and R01GM080139. S Shahmoradian was supported by a fellowship from the Nanobiology Interdisciplinary Graduate Training Program of the W M Keck Center for Interdisciplinary Bioscience Training of the Gulf Coast Consortia (NIH Grant No. T32EB009379). We thank Htet Khant for assistance with initial cryoEM screening and data collection.

## Additional information

### Funding

| Funder | Grant reference number | Author |
| --- | --- | --- |
| National Institutes of Health | PN2EY016525 | Sarah H Shahmoradian, Jesus G Galaz-Montoya, Judith Frydman, Steven J Ludtke, Wah Chiu |
| National Institutes of Health | P41GM103832 | Michael F Schmid, Steven J Ludtke, Wah Chiu |
| NIH (via W. M. Keck Center for Interdisciplinary Bioscience Training of the Gulf Coast Consortia) | T32EB009379 | Sarah H Shahmoradian |
| National Institutes of Health | R01GM080139 | Jesus G Galaz-Montoya, Steven J Ludtke |

The funders had no role in study design, data collection and interpretation, or the decision to submit the work for publication.

### Author contributions

SHS, Prepared aggregation reactions, Collected all 2D images for time series and huntingtin + TRiC tilt series, Initial tomogram reconstructions and annotation, Responsible for figures 1 and 9, Conception and design, Acquisition of data, Analysis and interpretation of data, Drafting and revising the text and all figures, Contributed unpublished essential data or reagents; JGG-M, Main development and testing of single particle tomography tools in EMAN2 used to process tomographic data, Performed tomographic reconstructions and annotation and all single particle tomography data processing and analysis of TRiC + huntingtin and TRiC control, Performed statistical analyses, Responsible for figures 2, 3, 4, 5, 6, 7 and 8, Conception and design, Analysis and interpretation of data, Drafting and revising the article, Contributed unpublished essential data or reagents; MFS, Advised single particle tomography tools development, and tomographic data processing, Conception and design, Drafting and revising the article; YC, Assisted with initial screening and 2D data collection, Acquisition of data, Analysis and interpretation of data; BM, Collected tilt series for control (TRiC with no huntingtin), Acquisition of data; CS, Purified TRiC and huntingtin construct, Assisted with aggregation reactions, Conception and design; JF, Conception and design, Drafting and revising the article; SJL, Development of single particle tomography tools in EMAN2

used to process tomographic data, Conception and design, Drafting and revising the article; WC, Conception and design, Analysis and interpretation of data, Drafting and revising the article

## Additional files

### Major dataset

The following datasets were generated:

| Author(s) | Year | Dataset title | Dataset ID and/or URL | Database, license, and accessibility information |
|---|---|---|---|---|
| Shahmoradian SH, Galaz JG, Schmid MF, Cong Y, Ma B, Spiess C, Frydman J, Ludtke SJ, Chiu W | 2013 | Single particle tomography of TRiC chaperonin incubated with substrate but not bound, "Cavity-empty TRiC" | EMD-5531; http://www.ebi.ac.uk/pdbe/entry/EMD-5531 | Publicly available at the Electron Microscopy Data Bank (http://www.ebi.ac.uk/pdbe/emdb/). |
| Shahmoradian SH, Galaz JG, Schmid MF, Cong Y, Ma B, Spiess C, Frydman J, Ludtke SJ, Chiu W | 2013 | Single particle tomography of TRiC chaperonin with internalized mhtt oligomers, "Cavity-occupied TRiC" | EMD-5530; http://www.ebi.ac.uk/pdbe/entry/EMD-5530 | Publicly available at the Electron Microscopy Data Bank (http://www.ebi.ac.uk/pdbe/emdb/). |
| Shahmoradian SH, Galaz JG, Schmid MF, Cong Y, Ma B, Spiess C, Frydman J, Ludtke SJ, Chiu W | 2013 | Single particle tomography of TRiC chaperonin with bound mhtt fibril, "Fibril-bound TRiC" | EMD-5532; http://www.ebi.ac.uk/pdbe/entry/EMD-5532 | Publicly available at the Electron Microscopy Data Bank (http://www.ebi.ac.uk/pdbe/emdb/). |
| Shahmoradian SH, Galaz JG, Schmid MF, Cong Y, Ma B, Spiess C, Frydman J, Ludtke SJ, Chiu W | 2013 | Single particle tomography of TRiC chaperonin with bound mhtt fibril, "Fibril-bound TRiC" (example 2) | EMD-5533; http://www.ebi.ac.uk/pdbe/entry/EMD-5533 | Publicly available at the Electron Microscopy Data Bank (http://www.ebi.ac.uk/pdbe/emdb/). |

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
