## [Decision Letter]

Thank you for sending your work entitled “TRiC’s Tricks Inhibit Huntingtin Aggregation” for consideration at *eLife*. Your article has been favorably evaluated by a Senior editor and 3 reviewers, one of whom, Werner Kühlbrandt, is a member of our Board of Reviewing Editors.

The Reviewing editor and the other reviewers discussed their comments before we reached this decision, and the Reviewing editor has assembled the following comments to help you prepare a revised submission.

Substantive concerns:

1) Although the authors state that the in vitro conditions employed in this work resemble those in vivo, it is not evident that a 1:5 molar ratio of TRiC to mhtt can be physiological, even if TRiC is one of the most abundant chaperonins in the cell. What would be typical in vivo levels of TRiC or mhtt? Please elaborate.

2) The number of glutamine residues in the pathogenic huntingtin exon 1 is typically 51–83. Which form is most abundant in cells? Why was the form with the lowest pathogenic Q frequency used? Is this form typical under physiological conditions?

3) One of the most interesting findings in the study is that a large portion of TRiC decorates the tip of the Htt fibril (instead of uniformly labeling the fibril). The implications of this finding in the context of mHtt fibril formation remain unclear.

4) TRiC was found at discrete locations along the fibrils. Can it be that many of the interactions between the TRiC and the fibrils were lost during sample preparation? How frequently was TRiC found at fibril ends and how many along fibrils? What was the ratio between free fibril ends/TRiC capped ends? Please provide some statistics.

5) After incubation with huntingtin, what proportion of TRiC was not interacting with the fibrils? Did these “unbound” complexes contain protein in their cavity ?

6) Was the map resolution really isotropic (as mentioned in the Methods section)?

7) The two TRiC rings in Figure 5 are very similar, and it is difficult to be sure that the difference between them is significant, as anyone knows who has ever correlated low-resolution EM maps. The smooth distribution of correlation factors in Figure 4 does not help, and the histograms of Figure 4 also do not indicate a clear difference between the two classes. Is there a quantitative measure to convince the reader that the difference between the two classes is real?

8) The difference density in Figure 8 looks nice, but would be even more convincing if the authors were to show a control with empty TRiCs processed by the same procedure alongside.

---

## [Author Response]

*1) Although the authors state that the* in vitro *conditions employed in this work resemble those* in vivo*, it is not evident that a 1:5 molar ratio of TRiC to mhtt can be physiological, even if TRiC is one of the most abundant chaperonins in the cell. What would be typical* in vivo *levels of TRiC or mhtt? Please elaborate*.

The mhtt concentration in our experiments was chosen to allow mhtt aggregation to occur in a reasonably short time without TRiC (31), and the ratio of TRiC to mhtt was chosen to yield initial-stage fibril aggregates and yet thwart large-aggregate growth (3). The TRiC:mhtt ratio used here is close to, but slightly higher than, normal physiological conditions. However, these can vary widely depending on many factors such as polyQ length, tissue of expression, and stage of the disease, etc. Furthermore, in the cell TRiC and htt (and mhtt) interact with additional partners, which would change the effective concentrations available for their interaction. The ratio we used in this study yields the same beneficial effects of TRiC over-expression in mhtt transfected cells, i.e., the growth of aggregates is inhibited. Importantly, the TRiC and mhtt conditions used here reproduce the chaperonin-mediated inhibition of mhtt aggregation observed in reconstituted systems, which is what we set out to characterize structurally. We have clarified this point in the revised text.

*2) The number of glutamine residues in the pathogenic huntingtin exon 1 is typically 51–83. Which form is most abundant in cells? Why was the form with the lowest pathogenic Q frequency used? Is this form typical under physiological conditions*?

TRiCs inhibition of mhtt aggregation has been biochemically characterized for several variants both in vitro and in vivo (3; 15; 35). Amongst these, mhttQ51 is of particular interest since mutations yielding a polyQ tract longer than ∼58Q are rare in patients (24) and N-terminal fragments of similar length (mhttQ53) have been isolated from brain tissue of patients (20). We include mention of these facts in the revised text.

*3) One of the most interesting findings in the study is that a large portion of TRiC decorates the tip of the Htt fibril (instead of uniformly labeling the fibril). The implications of this finding in the context of mHtt fibril formation remain unclear*.

From our tomograms, mhtt fiber-bundles, and thick fibers themselves are more tapered towards their end than in the middle (see Video 1). “Thick” mhtt *fibers* at later time points (4h post-initiation of aggregation onward) could be comprised of “thin” *fibrils*. Our data suggest that multiple, individual, thin fibril ends exposing the N17 motif known to bind TRiC might be abundantly available at the ends of thicker fibers, but only intermittently along the sides. Fibers might branch out and grow from these exposed tips (36) on their sides, or they might form lateral contacts through them with other fibers.

Our results imply that TRiC efficiently blocks fibril elongation by binding to the site of elongation, i.e. any exposed tips.

*4) TRiC was found at discrete locations along the fibrils. Can it be that many of the interactions between the TRiC and the fibrils were lost during sample preparation? How frequently was TRiC found at fibril ends and how many along fibrils? What was the ratio between free fibril ends/TRiC capped ends? Please provide some statistics*.

It is unlikely that interactions would be lost during sample preparation, since TRiC and mhtt were co-incubated immediately prior to initiating aggregation in vitro, as opposed to purified from cells. There was no additional sample preparation performed before plunge-freezing.

As reported in our manuscript, 37% of TRiCs picked as fibril-bound from our tomograms locate to discernible fibril tips; thus 63% bind elsewhere, but even those TRiCs not seen bound at the ends of the fibers might still be capping short stub-ends on fiber sides. We speculate that binding sites along the sides of fibers are “short tips”. To answer the last question, we observe ∼45% of distinguishable tips to be capped, but many tips (capped or uncapped) might be obscured by the dense tangles, or may be too short and budding from the side to be clearly discerned as tips.

Following the reviewers’ suggestion, we now make explicit mention of all of these statistics in our manuscript.

*5) After incubation with huntingtin, what proportion of TRiC was not interacting with the fibrils? Did these “unbound” complexes contain protein in their cavity* ?

We picked 326 TRiCs as fibril-bound and 890 as freestanding or “not discernibly bound to fibers”. Therefore, ∼27% of all extracted TRiCs (1216) seem to be interacting with fibers.

We were able to identify a sub-population of freestanding TRiCs that did in fact contain density in their cavity, as shown through Figures 4, 5 and 8. Cavity-occupied TRiCs account for ∼29% of all TRiC subvolumes (356 of 1216). We now mention these statistics explicitly in the manuscript.

*6) Was the map resolution really isotropic (as mentioned in the Methods section)*?

We made no claim of isotropic resolution in the Methods. Rather, we stated that tomograms display inherently anisotropic resolution and therefore it is challenging to perform conclusive analyses on the raw data without averaging.

Anisotropic resolution in cryoET can be attributed to factors such as the missing wedge and preferred orientation of the specimen. However, the Fourier transform of our averages shows no missing wedge, indicating that we averaged sufficient particles in different orientations. In fact, our freestanding TRiC maps agree well with the published cryoEM structure of apo-TRiC (Figure 7). Therefore, anisotropy doesn’t seem to pose particular concern in our freestanding TRiC averages.

On the other hand, because of their conformational heterogeneity, our fibril-bound averages are comprised of small numbers of subtomograms and are thus more prone to anisotropy in resolution.

*7) The two TRiC rings in Figure 5 are very similar, and it is difficult to be sure that the difference between them is significant, as anyone knows who has ever correlated low-resolution EM maps. The smooth distribution of correlation factors in Figure 4 does not help, and the histograms of Figure 4 also do not indicate a clear difference between the two classes. Is there a quantitative measure to convince the reader that the difference between the two classes is real*?

We agree that visually the structures in Figure 5 share similarities; they are, after all, the same specimen. The smooth distribution of correlation factors in Figure 4 could suggest heterogeneity, but is not sufficient to demonstrate it (it is necessary, but not sufficient). Even a conformationally homogenous specimen would show a similarly smooth distribution (but perhaps with a slower falloff) given the nature of cryoET data: 1) The raw data (individual TRiCs) are very noisy and 2) the missing wedge affects individual raw particles in different ways. Both of these factors introduce variations that contribute to a fall in correlation across the entire dataset. This curve, however, justifies the next step we took: because the curve was continuous, we performed statistical analyses to determine the significance of the differences between the structures shown in Figure 5. We divided the entire set of freestanding TRiCs into quintiles. The averages of the quintiles progressed from “highly correlating and (thus) completely empty” to “lower correlating and very full”. We decided to exclude the middle quintile to minimize ambiguities, and performed our statistical analyses using the top two quintiles (Figure 5) versus the two bottom quintiles (Figure 5). The t-values of our analyses indicate that the differences in mean-intensity (proportional to mean-mass) between the particles in each population are *very* statistically significant and, furthermore, that these differences reside in the cavity, and not in the TRiC chaperonin itself.

Having demonstrated that one population of freestanding TRiC contained statistically significant density inside (and that the other did not), we further processed it to localize the density within. We arrived at Figure 8, where the mass inside is found to be asymmetrically disposed within the cavity of TRiC.

*8) The difference density in Figure 8 looks nice, but would be even more convincing if the authors were to show a control with empty TRiCs processed by the same procedure alongside*.

We processed our control TRiC particles (never exposed to mhtt) in the same way as we processed freestanding TRiCs incubated with mhtt. We show their final average in Figure 6. In intermediate steps, there was no extra density lining any of our control averages. Furthermore, we were not able to find with statistical confidence two populations from our control TRiCs that were significantly different.

References

Behrends C, Langer CA, Boteva R, Böttcher UM, Stemp MJ, Schaffar G. 2006. Chaperonin TRiC promotes the assembly of polyQ expansion proteins into nontoxic oligomers. Mol Cell 23:887-897. doi:10.1016/j.molcel.2006.08.017.

Kitamura A, Kubota H, Pack CG, Matsumoto G, Hirayama S, Takahashi Y, Kimura H, Kinjo M, Morimoto RI, Nagata K. 2006. Cytosolic chaperonin prevents polyglutamine toxicity with altering the aggregation state. *Nat Cell Biol* 8:1163-170. doi:10.1038/ncb1478.

Lunkes, Lindenberg KS, Ben-Haïem L, Weber C, Devys D, Landwehrmeyer GB, Mandel JL, Trottier Y. 2002. Proteases acting on mutant huntingtin generate cleaved products that differentially build up cytoplasmic and nuclear inclusions. Molecular cell 10(2): 259-269.

Myers, Richard H. “Huntington's disease genetics.” NeuroRx 1.2 (2004): 255-262.

Scherzinger E, Sittler A, Schweiger K, Heiser V, Lurz R, Hasenbank R, Bates GP, Lehrach H, Wanker EE.“Self-assembly of polyglutamine-containing huntingtin fragments into amyloid-like fibrils: implications for Huntington’s disease pathology.” Proceedings of the National Academy of Sciences 96.8 (1999): 4604-4609.

Tam S, Geller R, Spiess C, Frydman J. 2006. The chaperonin TRiC controls polyglutamine aggregation and toxicity through subunit-specific interactions. Nat Cell Biol 8:1155-162. doi: 10.1038/ncb1477.

Tam S, Spiess C, Auyeung W, Joachimiak L, Chen B, Poirier MA, et al. 2009. The chaperonin TRiC blocks a huntingtin sequence element that promotes the conformational switch to aggregation. Nat Struct Mol Biol 16:1279-285. doi: 10.1038/nsmb.1700.